# Investigating the Use of a Liquid Immunogenic Fiducial Eluter Biomaterial in Cervical Cancer Treatment

**DOI:** 10.3390/cancers16061212

**Published:** 2024-03-20

**Authors:** Michele Moreau, Lensa S. Keno, Debarghya China, Serena Mao, Shahinur Acter, Gnagna Sy, Hamed Hooshangnejad, Kwok Fan Chow, Erno Sajo, Jacques Walker, Philmo Oh, Eric Broyles, Kai Ding, Akila Viswanathan, Wilfred Ngwa

**Affiliations:** 1Department of Radiation Oncology and Molecular Radiation Sciences, Department of Biomedical Engineering, Johns Hopkins Hospital, Baltimore, MD 21287, USA; lkeno1@jh.edu (L.S.K.); dchina1@jh.edu (D.C.); smao9@jhmi.edu (S.M.); sacter1@jh.edu (S.A.); gsy1@jh.edu (G.S.); hamed@jhu.edu (H.H.); kding1@jhmi.edu (K.D.); aviswan3@jh.edu (A.V.); 2Department of Chemistry and Department of Physics (Medical Physics), University of Massachusetts Lowell, Lowell, MA 01854, USA; kwokfan_chow@uml.edu (K.F.C.); erno_sajo@uml.edu (E.S.); 3Department of Health Administration and Human Resources, The University of Scranton, 800 Linden Street, Scranton, PA 18510, USA; 4Nanocan Therapeutics Corporation, Princeton, NJ 08540, USA; jwalker@nanocan.life (J.W.); ebroyles@nanocan.life (E.B.)

**Keywords:** image-guided radiotherapy, liquid fiducials, smart radiotherapy biomaterials, cervical cancer, anti-CD40, toxicities

## Abstract

**Simple Summary:**

Current treatment options for advanced cervical cancer are limited, with a 5-year overall survival rate of 17%. New effective therapeutic approaches that can increase survival and reduce disparities are greatly needed. In this study, an approach using a multifunctional liquid immunogenic fiducial eluter (LIFE) that can provide image contrast and enhance therapeutic efficacy during hypo-fractionated radiotherapy is investigated. We found this technique provides both computed tomography (CT) and magnetic resonance imaging (MRI) image contrast to guide radiotherapy. Additionally, in a model of metastatic cervical cancer, local treatment with LIFE and radiotherapy resulted in distant disease control and improvement in survival. Successful development of this approach may provide a short therapeutic option that could enhance treatment outcomes and reduce disparities. LIFE Biomaterial may also significantly provide a cheaper and convenient alternative to prolonged systemic treatment with opportunities to increase access to care and reduce health outcome disparities.

**Abstract:**

Globally, cervical cancer is the fourth leading cancer among women and is dominant in resource-poor settings in its occurrence and mortality. This study focuses on developing liquid immunogenic fiducial eluter (LIFE) Biomaterial with components that include biodegradable polymers, nanoparticles, and an immunoadjuvant. LIFE Biomaterial is designed to provide image guidance during radiotherapy similar to clinically used liquid fiducials while enhancing therapeutic efficacy for advanced cervical cancer. C57BL6 mice were used to grow subcutaneous tumors on bilateral flanks. The tumor on one flank was then treated using LIFE Biomaterial prepared with the immunoadjuvant anti-CD40, with/without radiotherapy at 6 Gy. Computed tomography (CT) and magnetic resonance (MR) imaging visibility were also evaluated in human cadavers. A pharmacodynamics study was also conducted to assess the safety of LIFE Biomaterial in healthy C57BL6 female mice. Results showed that LIFE Biomaterial could provide both CT and MR imaging contrast over time. Inhibition in tumor growth and prolonged significant survival (* *p* < 0.05) were consistently observed for groups treated with the combination of radiotherapy and LIFE Biomaterial, highlighting the potential for this strategy. Minimal toxicity was observed for healthy mice treated with LIFE Biomaterial with/without anti-CD40 in comparison to non-treated cohorts. The results demonstrate promise for the further development and clinical translation of this approach to enhance the survival and quality of life of patients with advanced cervical cancer.

## 1. Introduction

Cervical cancer is the fourth leading cancer among women worldwide [1,2]. Most cervical cancers are driven by the human papillomavirus (HPV). The three pillars for eliminating cervical cancer include HPV vaccination, screening, and treatment [3,4,5,6]. The highest burden of cervical cancer incidence and mortality is in low-resource settings in the USA and low- to middle-income countries [7,8,9].

Common treatment options for patients with cervical cancer include surgery if diagnosed in its early stages [10,11,12]. Radiation therapy and chemotherapy, separately or in combination, are used for patients with advanced stages of cervical cancer [13,14,15,16]. External beam and internally delivered radiation (brachytherapy) are often used in the radiotherapy of cervical cancer [17,18]. Newer treatments, such as targeted therapy and immunotherapy, have been used in the last decades, including current FDA approval of the combination of pembrolizumab and chemoradiotherapy for the treatment of newly diagnosed, high-risk, locally advanced cervical cancer [19,20,21,22,23,24]. However, limitations of these approaches include long treatment regimens, side effects, and high costs that drive persistent disparities and impact patient’s overall quality of life [25,26,27,28]. New effective treatment options that can also reduce disparities are needed, especially for patients with advanced cervical cancer.

This study investigates a novel therapeutic approach involving liquid immunogenic fiducial eluter (LIFE) Biomaterial loaded with immunotherapeutic drug anti-CD40 monoclonal antibody. Anti-CD40 has been chosen specifically in this study because it has been shown in previous studies to be an effective immunoadjuvant in enhancing in situ vaccination during radiotherapy [29,30]. One of the treatment cohorts explored if adding anti-PD1 would have a significant additional impact on the radiotherapy plus LIFE Biomaterial treatment. Therapeutic approaches involving drug-loaded systems have been investigated pre-clinically, including hydrogel delivery systems that are biocompatible, non-toxic, and can provide space–time control over the release, localization, or distribution of various drugs. This takes advantage of the ability to optimize or control their behavior, biodegradation, and capacity to protect labile drug payloads from degradation [31,32,33,34]. Meanwhile, previous work has investigated novel fiducials for use during cervical cancer radiotherapy. For example, a calcium phosphate cement (CPC) marker was investigated as an injectable non-metallic fiducial marker [35]. This study, conducted in six patients receiving 3–5 injections of the CPC markers, demonstrated that such fiducials injected into cervical tumors can be visualized on cone beam computed tomography (CT) and MRI with reductions in marker loss and artifacts [33]. In this study, LIFE Biomaterial is designed to accomplish more in serving as multifunctional technology with the capacity to not only load drugs but also provide CT and MRI image guidance during radiotherapy. This study explores LIFE biomaterial’s potential to (1) provide image guidance during radiotherapy similar to fiducial markers with investigations in both animals and cadavers, (2) enhance therapeutic efficacy in advanced cervical cancer animal models, and (3) ensure minimal toxicity as investigated in healthy mice.

## 2. Materials and Methods

### 2.1. Preparation of LIFE Biomaterial

LIFE Biomaterial was designed with a combination of natural polymers, namely 2% (*w*/*v*) of chitosan (Sigma Aldrich, St. Louis, MO, USA) and 4% (*w*/*v*) of sodium alginate (Sigma Aldrich, St. Louis, MO, USA) hydrogel, mixed in a 1:1 ratio. In addition, two solutions of nanoparticles were prepared separately. One of the prepared solutions was 0.85 g/mL of titanium dioxide (TiO_2_ Anatase, 99.5%, 5 nm, US Research Nanomaterials Inc., Houston, TX, USA). The other solution was 0.287 g/mL of Omniscan gadodiamide gadolinium-based (GdNPs) nanoparticles (GE Healthcare, Silver Spring, MD, USA). Both the TiO_2_ and the GdNPs were mixed in a 1:1 ratio and added to the collective LIFE Biomaterial hydrogel in a ¼:1 ratio, respectively. LIFE Biomaterial hydrogel was maintained at 4 °C until treatment. On treatment day, LIFE Biomaterial was partitioned based on the number of mice being treated, and 20 µg per mouse of anti-CD40 (Clone FGK4.5, BioXCell, Lebanon, NH, USA) antibody was added to the volume of hydrogel to constitute LIFE Biomaterial. Among the treatment groups, one group received an intraperitoneal injection of 200 µg of anti-PD1 (Clone RMP1-14, BioXCell, Lebanon, NH, USA), in addition to the intratumoral injection of LIFE Biomaterial loaded with 20 µg of anti-CD40 and exposed to one fraction of 6 Gy. In this study, LIFE Biomaterial was injected intratumorally in one subcutaneous cervical tumor of a mouse to assess both the image guidance during radiotherapy and the therapeutic potential. The same amount was injected subcutaneously in healthy non-tumor-bearing mice to assess any toxicity-related effects.

The LIFE Biomaterial schematic is highlighted in Figure 1 with nanoparticles and liquid/drug localized in a 3D network of cross-linked polymer chains that is easy to administer to tumors similar to fiducials. A physical depiction of the LIFE Biomaterial is displayed in Figure 2A.Scanning and transmission electron microscopy (SEM and TEM, respectively) images highlighting the morphology and surface texture of LIFE Biomaterial colloid are shown in Figure 2B,C.

### 2.2. TC1 Cell Culture Preparation and Tumor Growth in Mice

A cervical cancer cell line, TC1 (ATCC^®^ CRL-2785, Manassas, VA, USA), was used to grow subcutaneous tumors in C57BL6 female mice following previously published protocols [34]. TC1 cells were cultured in RPMI 1640 medium supplemented with 2 Mm L-glutamine adjusted to contain 1.5 g/L sodium bicarbonate, 4.5 g/L glucose, 10 mM HEPES, 1.0 mM sodium pyruvate, 0.1 mM non-essential amino acids, and 10% fetal bovine serum. All cells were cultured at 37 °C in a humidified incubator with 5% CO_2_. Immunocompetent wild-type C57BL/6 strain female mice were acquired from Charles River at 6–8 weeks old. They were inoculated subcutaneously with 3 × 10^5^ cells mixed with Matrigel in the lateral flanks of mice and allowed to grow to a palpable tumor size of approximately 3 mm in diameter within ten days before the start of treatment. Mice tumor volume was determined using the formula V = 0.5 × length × (width^2^). A digital caliper was used to measure the longitude protrusion of the tumor designated as the tumor length and the latitude projection as the tumor width. Animal experiments followed the guidelines and regulations of the Johns Hopkins University Animal Care and Use Committee (ACUC) set under protocol# MO21M281. Mice maintenance in the Johns Hopkins University animal facility was according to the Institutional Animal Care and Use Committee approved guidelines.

### 2.3. Imaging and Efficacy Study

LIFE Biomaterial formulations prepared as described above were used to assess CT and MRI image contrasts in the C57BL6 mice bearing cervical subcutaneous tumors. Images were collected over different time points. The imaging was performed over several weeks to establish the potential for providing imaging contrast for hypo-fractionated radiotherapy. To further assess the feasibility of LIFE Biomaterial in providing imaging, contrast imaging was also conducted in a human cadaver approved by the Johns Hopkins Institutional Review Board (IRB) for protocol number NA_00070589 (PI KD). Refrigerated, unfixed cadaveric specimens were injected with LIFE Biomaterial following CT simulation (TOSHIBA Helical CT scan with 2 mm slice thickness, 120 kVp, and X-ray tube current of 100 mA) with the cadaveric specimen in the supine position. The specimens were then imaged again. For MRI, a Philips Achieva 3.0 T MRI System with BODY Transmit Coil was used with a repetition time of 5.31 ms, a flip angle of 10^0^, a percent phase field of view of 70.833, and a slice thickness of 0.9 mm.

The potential of the formulations to enhance treatment outcomes was also investigated in the same mice bearing cervical tumors. The investigation with more than one tumor per mouse was representative of subjects with advanced cervical cancer burden with more than one tumor, as in the case of metastasis. A small animal radiation research platform was used to deliver a 6 Gy dose at 220 kVp locally at the tumor site using a 10 × 10 mm collimator size. Tumor volume and survival were assessed over time. Survival analysis using the log-rank (Mantel-Cox) test was performed using GraphPad Prism version 9.5.1 for Windows. (GraphPad Software, San Diego, CA USA). Analyzed data were deemed significant if their *p*-values were within the following ranges: (* *p* < 0.05) and (** *p* < 0.01).

### 2.4. Pharmacodynamics of LIFE Biomaterial in Healthy Mice

Seven- to eight-week-old wild-type C57BL6 female mice were acquired from Charles River. Eleven- to thirteen-week-old healthy mice (*n* = 36) were allocated into several groups. Animals in one cluster received one inoculation of LIFE Biomaterial loaded with anti-CD40 (20 µg) (*n* = 12). A second set was administered with LIFE Biomaterial without anti-CD40 (*n* = 12). These noted groups were investigated for comparison against a healthy animal group with no treatment (*n* = 12). Mice were maintained, and all studies followed the Johns Hopkins University ACUC-approved standard of practice and protocol (#MO21M281).

Blood was drawn from the vena cava vein at several time points (day 1, 30, 60, and 90) post-treatment to evaluate hepatic and renal function parameters, including alkaline phosphatase, total protein, alanine transaminase (ALT), aspartate aminotransferase (AST), bilirubin direct, total bilirubin, gamma-glutamyl transferase (GGT), glucose, blood urea nitrogen (BUN), and creatinine, among other parameters. Results were compared to those of healthy mice with no treatment. Blood serum was separated within one hour after collection by centrifuging collected blood at 5000 rpm for 10 min at 4 °C following a standard protocol and sent for analysis. All collected sera were sent to VRL Laboratories (San Antonio, TX, USA) for analysis.

Complete blood count (CBC) hematology analysis was performed for a variety of parameters, highlighting red blood cells (RBCs), Hematocrit (HCT %), mean corpuscular volume (MCV), mean corpuscular hemoglobin (MCH), monocyte (MONO) count, and hemoglobin (HGB), among other CBC parameters at the following time points: days 1, 30, 60, and 90 post-treatment following vena cava blood sampling. Whole blood was collected into EDTA tubes and kept at 2–8 °C to send for analysis. Whole blood samples were sent for analysis to VRL Laboratories (San Antonio, TX, USA), within 24 h after collection.

Histopathological analysis of different organs (e.g., heart, lung, left and right kidneys, liver, and spleen) was achieved following vena cava blood sampling of treated and non-treated healthy mice at different time points post-treatment: days 1, 30, 60, and 90. Tissue samples were sent to VRL Laboratories (San Antonio, TX, USA) for histopathology assessments.

## 3. Results

### 3.1. Potential of LIFE Biomaterial for Image-Guided Radiotherapy

The first studies assessed the potential of LIFE Biomaterial to provide imaging contrast during radiotherapy similar to fiducials. Figure 3a shows both CT and MR imaging contrast over time from LIFE Biomaterial, as indicated by the red arrow. The region with the secondary tumor that was not administered with LIFE Biomaterial is highlighted within the yellow dotted circle. CT images are shown in Figure 3b starting a week post-treatment up to 23 days post-administration of LIFE Biomaterial. Figure 3c highlights both imaging modalities of LIFE Biomaterial in human cadavers.

### 3.2. Efficacy of the LIFE Biomaterial in Cervical Cancer

Two separate studies were conducted. In the first study, thirty mice bearing two subcutaneous tumors were used and divided into the following cohorts: (a) control (no treatment) (*n* =7); (b) one fraction of 6 Gy (*n* = 7); (c) LIFE Biomaterial loaded with 20 µg of anti-CD40 antibody (*n* = 8); and (d) LIFE Biomaterial loaded with 20 µg of anti-CD40 antibody and receiving one fraction of 6 Gy (*n* = 8). During the second study, the treatment parameters were varied with the following groups: (a) control (no treatment) (*n* = 6); (b) LIFE Biomaterial loaded with 20 µg of anti-CD40 antibody and 6 Gy (*n* = 6); (c) LIFE Biomaterial loaded with 20 µg of anti-CD40 antibody in addition to free injection of 20 µg of anti-CD40 and 6 Gy (*n* = 6); (d) LIFE Biomaterial loaded with 20 µg of anti-CD40 antibody in addition to intraperitoneal injection of 200 µg of anti-PD1 and 6 Gy (*n* = 6); and (e) free injection of 20 µg of anti-CD40 intratumorally. All treatments were administered in only one primary tumor per mouse, not the secondary tumor (Figure 4a). Both tumors were monitored for tumor volume and mice survival over time.

Beyond LIFE Biomaterial providing image contrast/guidance (as in Figure 3), LIFE Biomaterial is designed to also augment the response to local therapy. Figure 4 highlights the inhibition observed for the groups locally treated with LIFE Biomaterial loaded with anti-CD40 monoclonal antibodies (mAbs) with/without one fraction of 6 Gy. Tumor growth was inhibited for both the treated/primary tumor and the untreated secondary tumor in both sets of studies shown in Figure 4b–e. Response in treatment groups without radiation highlights the fact, also seen in other studies, that anti-CD40 itself can cause immunogenic cell death and in situ vaccination [35,36].

Other groups that received either a free injection of anti-CD40 or one fraction of 6 Gy alone also showed some tumor growth inhibition but not as much as with LIFE Biomaterial treatment groups. Overall, from these two studies, a significant increase in survival is consistently observed for groups treated with LIFE Biomaterial. The results also show the potential for further optimization studies, e.g., with other doses of anti-CD40 or radiotherapy.

### 3.3. Pharmacodynamics of LIFE Biomaterial in Healthy Female Mice

It is important to show that LIFE Biomaterial itself is not toxic to healthy animals. Immunocompetent, wild-type C57BL/6 female mice were used for toxicity studies in healthy mice. Biochemical analysis evaluated several parameters related to renal and hepatic functions highlighted in Figure 5 and Appendix A following vena cava blood collection techniques comparing the no-treatment group against the treated groups (*n* = 3–4/time point). The time points were days 1, 30, 60, and 90 post-treatment.

#### 3.3.1. Hepatic Function

Hepatic function was evaluated in C57BL6 female mice comparing a healthy group of mice against the treated cohorts using LIFE Biomaterial unloaded/loaded with anti-CD40 monoclonal antibody. Various parameters (e.g., alanine transaminase (ALT), aspartate aminotransferase (AST), total bilirubin, gamma-glutamyl transferase (GGT), bilirubin direct, and alkaline phosphatase) were tested to assess the effect of the drug product on the liver. The toxicity parameters for both treatment and control groups were found to be within the range that shows minimal toxicity at the different time points tested, as highlighted in Figure 5a. All the parameters measured for hepatic function are mentioned in Appendix A.

#### 3.3.2. Renal Function

Renal function was also gaged where different parameters (e.g., glucose, blood urea nitrogen (BUN), total protein, and creatinine) were tested to assess how safe the drug product is as it relates to impacting the kidney function of healthy mice. Based on the reference ranges provided by VRL Laboratories, there was no adverse effect on kidney function, as emphasized in Figure 5b. All the parameters measured for renal functions are displayed in Appendix A.

#### 3.3.3. Cell Blood Count Analysis

Cell blood count analysis was performed using whole blood collected from healthy C57BL6 female mice at different time points (days 1, 30, 60, and 90) to investigate the overall health and find a wide range of conditions, including anemia, infection, and other health conditions after administering LIFE Biomaterial treatment with/without loaded anti-CD40. Hematological analysis was performed for many parameters, such as red blood cells (RBCs), hemoglobin (HGB), monocyte (MONO) count, hematocrit (HCT %), mean corpuscular hemoglobin (MCH), and mean corpuscular volume (MCV), among other parameters. Figure 5c displays the normal blood cell count for the treated groups compared to the non-treated group of mice. Appendix A displays the complete blood cell count parameters for all the cohorts at all collection time points.

This study assessed biochemical parameters related to renal and hepatic functions of healthy female mice treated with LIFE Biomaterial unloaded/loaded with anti-CD40 mAbs and compared the results with a control (no treatment) healthy cohort of mice at different time points. Figure 5a,b and Appendix A highlight minimum to no damages incurred by the liver and/or kidneys among all the cohorts at days 1, 30, 60, and 90 post-treatment. The average values represented on the plots were within the reference range given for each tested parameter by VRL Laboratories or demonstrated minimal to no toxicities. Comparable cell blood count values for control and treated groups are shown in Figure 5c and Appendix A, as observed for all the cohorts at the different collection times.

Histopathological assessments of the organs collected (heart, lung, liver, left and right kidneys, and spleen) at each collection time point (days 1, 30, 60, and 90) post-treatment are reported in Appendix A, showing minimal to no lesions found in all the organs, except for the liver, which incurred minimal to mild lesions that were comparable across all the tested cohorts. A lesion is defined as any damage that occurred due to any treatments given to the mouse. Some common liver lesions that occurred in either the no-treatment or the treated cohorts consisted of (a) micro-granulomas—small aggregates of inflammatory cells, up to 100 cells, consisting of macrophages, lymphocytes, and neutrophils, which often contain one or more entrapped degenerating hepatocytes. In micro-granulomas, the mononuclear inflammatory cells predominate the cellular aggregate. (b) Micro-abscesses are small aggregates of inflammatory cells, up to 100 cells, consisting of macrophages, lymphocytes, and neutrophils, which often contain one or more entrapped degenerating hepatocytes. In micro-abscesses, the neutrophils are the predominant inflammatory cells in the cellular aggregate. Appendix A provide a detailed account of the pathology reports performed for each organ at each collection time point for all the cohorts using a scoring definitions system defined by VRL Laboratories as 0 = No finding; 1 = Minimal; 2 = Mild; 3 = Moderate; 4 = Marked; 5 = Severe; N = Normal; M = Missing; MF = Multifocal; F = Focal; D = Diffuse; U = Unilateral; and B = Bilateral. Minimum to no significant lesions were reported in Appendix A for all the organs collected for both the non-treated group and the treated cohorts.

## 4. Discussion

The results in this study demonstrate the potential for the use of LIFE Biomaterial technology for imaging contrast over time similar to that of fiducial markers. Development and clinical translation could provide a new option for fiducial markers. A limitation of this study is that the time point LIFE Biomaterial biodegrades and ceases to provide imaging contrast has not been established. However, ongoing studies will complete this characterization. The current studies clearly demonstrate image contrast that can be useful for image-guided radiotherapy accommodating hypo-fractionated radiotherapy schedules. Importantly, this provides contrast for both MRI and CT, as treatment planning for cervical cancer often incorporates both modalities. Notably, LIFE Biomaterial is composed of natural biodegradable polymers and biocompatible nanoparticles that can be cleared over time.

The advantage of LIFE Biomaterial is that it is multifunctional and can also enhance therapeutic efficacy, as seen in this experimental model for metastatic cervical cancer. Limitations of this study in this respect include the need for the variation of additional parameters, such as the anti-CD40 dose or radiotherapy dose, including other hypo-fractionation schedules, to provide more insights and optimization outcomes. The studies demonstrate the potential for the optimization of therapy outcomes via this approach. Ongoing studies will explore these parameters and immune cell populations to elucidate the apparent response of the non-treated tumors, consistent with in situ vaccination reported in previous work. The results also motivate future studies that could explore the release of cytokines and any surface expression of different inhibitory/stimulatory molecules. Another limitation of this study in terms of therapy outcomes consists of using the subcutaneous model of cervical cancer for its simplicity. Studies involving orthotopic or genetic models would be valuable in providing additional rigor.

CD40, a receptor from the tumor necrosis factor (TNF) family, is heavily expressed on dendritic cells, which are vital for inciting an antigen-specific T-cell response [37,38]. In addition, cancer cells can undergo apoptosis induced by CD40 intracellular signaling, generating neo-antigens to help boost anti-cancer immune responses [39,40,41]. This ability is highlighted by the results showing that the use of anti-CD40/LIFE Biomaterial without radiotherapy can also lead to slowing both treated and untreated tumors in the same animal. In perspective, previous work [42,43] notes that the quality of CD40 stimulation may vary as the CD40 receptor has pleiotropic effects, and different residues on the CD40L/CD40 interaction may result in the differential production of pro-inflammatory cytokines. Our previous work [37,44] highlighted that when engaged by CD40L or by an agonistic antibody, CD40 signaling may lead to NF-κB upregulation and the expression of costimulatory ligands, the production of IL-12 and other cytokines, enhanced antigen presentation, and in the case of dendritic cells, the upregulation of CCR7 and trafficking to the draining lymph node. Overall, this previous work suggests the potential for synergies between radiotherapy and adjuvants like anti-CD40. One of the major limitations of anti-CD40 immunotherapy has been systemic toxicity. The use of LIFE Biomaterial presents advantages to minimize the off-target toxicities that can impact the quality of the treatment and life of subjects [45]. The addition of an anti-PD1 antibody did not seem to generate any significant additional effect on the tumor volume regression and mice survival over time. Further studies of varying parameters, such as dose, are needed to clearly establish any potential additional benefits.

The pharmacodynamics study was conducted at pre-determined time points for healthy cohorts of mice comparing a non-treated group to those treated with one single injection of either unloaded LIFE Biomaterial or LIFE Biomaterial loaded with anti-CD40 monoclonal antibody. Hepatic and renal parameter levels from mice treated with LIFE Biomaterial unloaded/loaded with anti-CD40 were found to be within the normal range and comparable with the levels from the untreated mice on days 1, 30, 60, and 90 post-intervention of the study. Similarly, the hematological analysis assessing the blood count levels remained within the normal range for the parameters tested with no substantial variations in the documented levels between the treatment groups and the control. The pathology report presented in the Appendix A further corroborates the minimal lesions results found in the collected organs, such as the heart, spleen, liver, left and right kidneys, and the lungs of the non-treated and treated cohorts. In studies further optimizing therapeutic efficacy, the toxicity parameters examined here can also be further investigated.

## 5. Conclusions

Overall, this study highlights pre-clinical evidence for a novel multifunctional radiotherapy biomaterial approach that can be developed for advanced cervical cancer treatment. The imaging, therapy, and safety results demonstrate that this approach has significant potential for optimization and clinical translation to enhance the survival and quality of life of advanced cervical cancer patients. Moreover, the potential for reducing treatment time/costs via such an approach provides an exciting opportunity for reducing disparities.

## Figures and Tables

**Figure 1 cancers-16-01212-f001:**
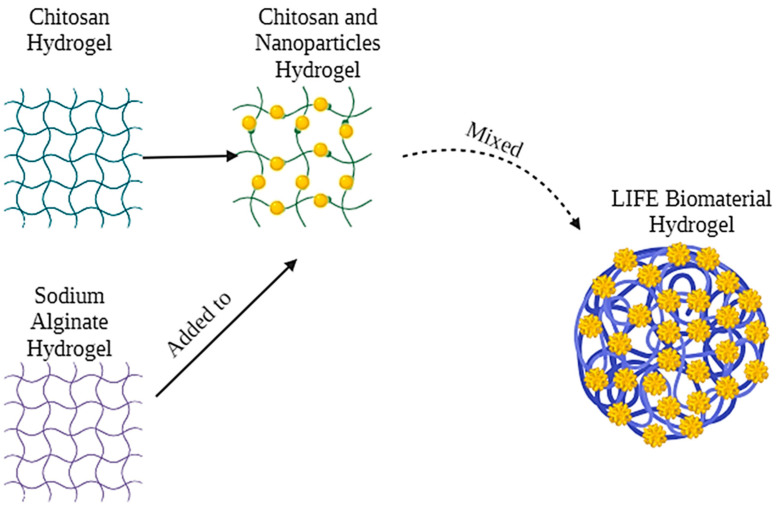
LIFE Biomaterial depiction highlights the mixture of two natural polymers consisting of chitosan and sodium alginate and also incorporating nanoparticles that can provide both CT and MRI imaging contrasts to guide radiotherapy and enhance therapy outcomes.

**Figure 2 cancers-16-01212-f002:**
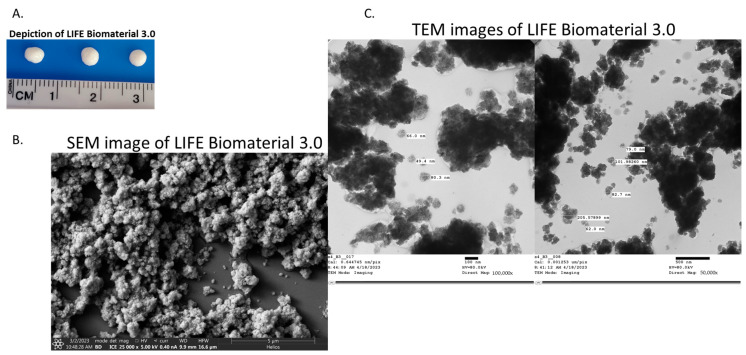
Characterization of LIFE Biomaterial formulated with titanium dioxide and omniscan gadodiamide nanoparticles. (**A**) Photographs of LIFE Biomaterial colloid/gel. (**B**,**C**) SEM and TEM images of LIFE Biomaterial colloid. These images also visualize the micro-architectural properties of LIFE Biomaterial, such as its surface topology and organization of the polymeric networks.

**Figure 3 cancers-16-01212-f003:**
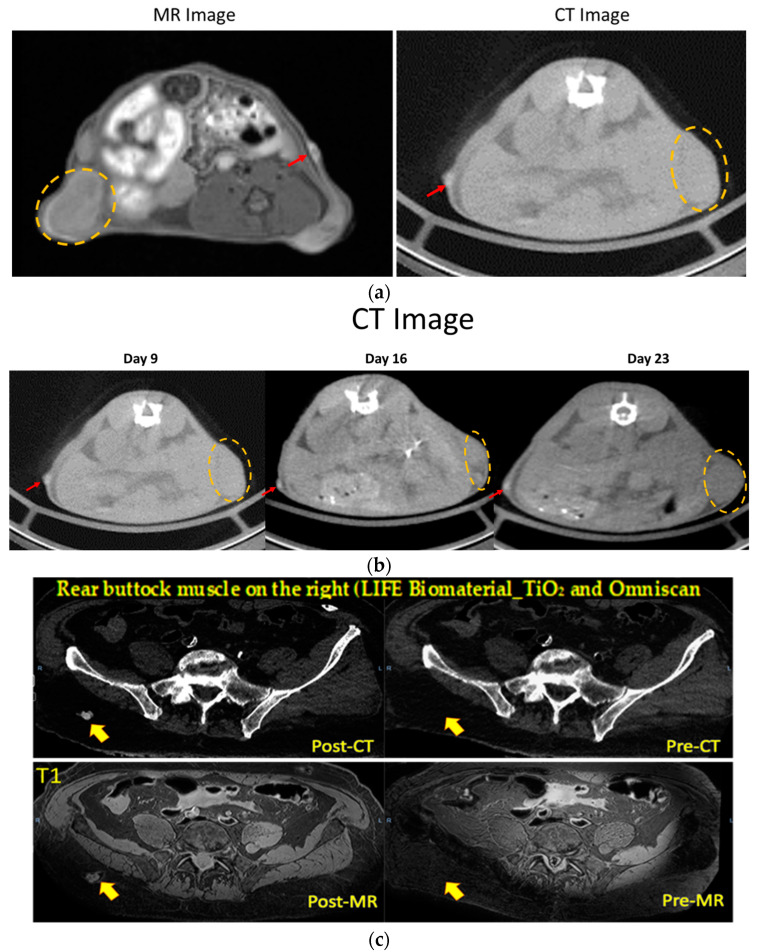
The image contrast of LIFE Biomaterial in cervical tumors and a human cadaver. (**a**,**b**) LIFE Biomaterial provided CT and MR contrasts as indicated by the red arrows within the same cervical tumor mouse where the tumor is highlighted by the orange dotted circles. The CT contrast can be observed up to day 23. (**c**) LIFE Biomaterial seen in both CT and MR imaging modalities in a human cadaver denoted by the red/orange arrows.

**Figure 4 cancers-16-01212-f004:**
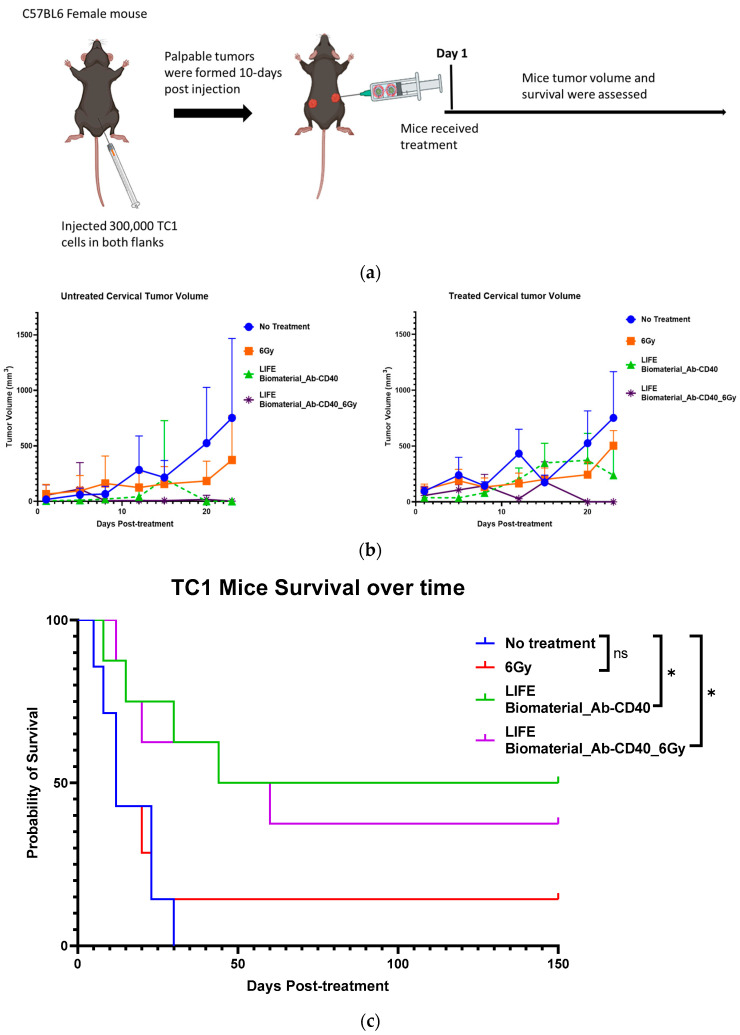
Efficacy of LIFE Biomaterial loaded anti-CD40 antibody in mice cervical tumors. (**a**) Study design using C57BL6 female mice bearing subcutaneous tumors on lateral flanks with treatment according to the groups shown in the plots in (**b**,**c**). Slower tumor growth was observed for the groups treated with LIFE Biomaterial loaded anti-CD40 antibody, and significantly (* *p* < 0.05) longer mice survival was observed for up to 5 months post-treatment. The following study also showed similar trends relating to slowed tumor growth (**d**) and significantly (* *p* < 0.05 and ** *p* < 0.01)) longer survival post-treatment and ‘ns’ denotes not significant (**e**), highlighting the potential to further optimize the treatment approach.

**Figure 5 cancers-16-01212-f005:**
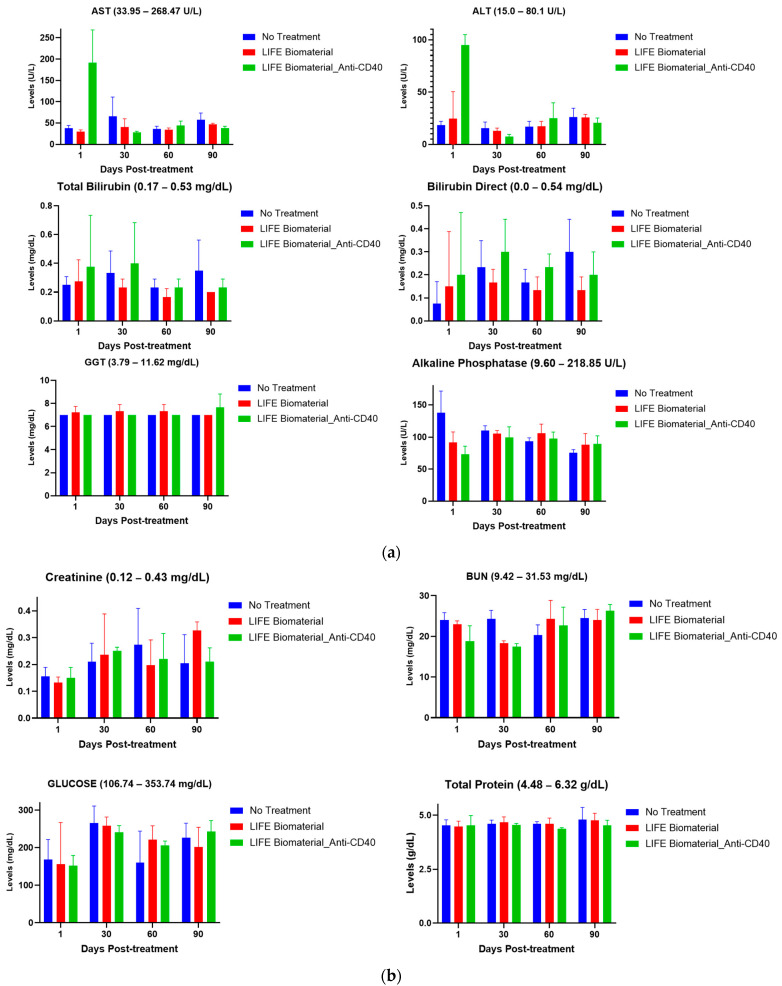
Pharmacodynamics of healthy mice comparing treated mice with LIFE Biomaterial unloaded/loaded with anti-CD40 against a healthy non-treated cohort of mice at different time points. The reference range for each parameter is annotated in the title of each plot. (**a**) Hepatic function parameters, such as alanine transaminase (ALT), aspartate aminotransferase (AST), bilirubin direct, total bilirubin, gamma-glutamyl transferase (GGT), and alkaline phosphatase, were within the normal reference range for all of the cohorts investigated. (**b**) Renal function parameters, including blood urea nitrogen (BUN), creatinine, total protein, and glucose, were all within the noted reference ranges given for each particular parameter for all of the tested cohorts. (**c**) Blood cell count analysis of red blood cells (RBCs), hemoglobin (HGB), hematocrit (HCT), mean corpuscular volume (MCV), mean corpuscular hemoglobin (MCH), and monocyte (MONO) count were within the reference range given by VRL Laboratories and were comparable to the healthy group of mice at the different time points tested.

## Data Availability

Data are contained within the article or Appendix A. Data are also available upon request from the corresponding author.

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
