# Peer review of "Investigating the Use of a Liquid Immunogenic Fiducial Eluter Biomaterial in Cervical Cancer Treatment"

_cancers, 2024, doi:10.3390/cancers16061212_

Round 1
Reviewer 1 Report
Comments and Suggestions for Authors
This is an excellent study with proper experimental design. The authors focus on a novel Liquid Immunogenic Fiducial Eluter (LIFE), designed to help in image-guided radiotherapy for advanced cervical cancer. The C57BL6 tumors in mice were treated with LIFE Biomaterial combined with radiotherapy. The visibility of the biomaterial was assessed through CT and MR imaging in human cadavers. Results indicated that the LIFE Biomaterial provided sustained imaging contrast and, when combined with radiotherapy, demonstrated inhibited tumor growth and prolonged survival, without toxicity.
Author Response
Thank you for reviewing our manuscript.
Reviewer 2 Report
Comments and Suggestions for Authors
the article can be published in the current form
Author Response
Thank you for your approval of the manuscript.
Reviewer 3 Report
Comments and Suggestions for Authors
The manuscript in case investigates a fiducial biomarker for cervical cancer treatment with proposed benefits in both tumor visualization and treatment response. It is the result of a complex teamwork which indeed highlights the preclinical efficiency of a novel biomaterial. Tumor injecting radiosensitizers have been already studied for a few years now, especially in soft tissue sarcomas; this study employs additional imaging and immunogenic benefits .
A few issues need to be clarified:
- line 35: please provide short explanation for the use of anti-cd40
- line 42: have the abstract clearly describe LIFE Biomaterial. Should we understand anti-cd40 are normally included or not ?
- line 68: provide reason for choosing anti-cd40 monoclonal antibody
- verify all references for common format. e.g. reference 33 authors written with capital letters
Author Response
The manuscript in case investigates a fiducial biomarker for cervical cancer treatment with proposed benefits in both tumor visualization and treatment response. It is the result of a complex teamwork which indeed highlights the preclinical efficiency of a novel biomaterial. Tumor injecting radiosensitizers have been already studied for a few years now, especially in soft tissue sarcomas; this study employs additional imaging and immunogenic benefits.
A few issues need to be clarified:
- line 35: please provide short explanation for the use of anti-cd40
A short phrase describing that anti-CD40’s use is as an immunoadjuvant has been added in Line 36 - 67.
- line 42: have the abstract clearly describe LIFE Biomaterial. Should we understand anti-cd40 are normally included or not ?
A sentence in Line 32 - 33 has been added to describe that the LIFE Biomaterial includes the immunoadjuvant anti-CD40.
- line 68: provide reason for choosing anti-cd40 monoclonal antibody
A sentence has been added to provide the reason for choosing anti-CD40 monoclonal antibody in Lines 71-74.
- verify all references for common format. e.g. reference 33 authors written with capital letters
All references have been verified.
Reviewer 4 Report
Comments and Suggestions for Authors
The manuscript entitled “Investigating the Use of a Liquid Immunogenic Fiducial Eluter Biomaterial in Cervical Cancer Treatment” submitted by Moreau et.al. discuss the use of a novel multifunctional radiotherapy biomaterial like liquid immunogenic fiducial eluter for the treatment of cervical cancer during radiotherapy. This technique makes use of both the CT and MRI which helps in the guided therapy. The authors showed that the use of this technology helps in controlling the cervical cancer also increasing the survival rates.
The authors have a clear hypothesis and to test this the author used appropriate experimental design. The manuscript is well written, and the statistical methods used are appropriate to the best of my knowledge. The material and methods sections provide complete information. The discussion is well written, highlighting the limitations of the study and shows the future perspective. However, there are some concerns as follows.
The authors need to mention the exact clone of anti-CD40 antibody used as there are different clones available in the market and how was this generated if they are generating it inhouse under the material and methods section.
What was the purpose of using the anti-PD1 antibody as the survival rates were almost same in this group and the CD40 treated group? Also, the authors need to mention the anti-PD1 antibody clone used.
The authors have not tested for the release of the cytokines and surface expression of the different inhibitory / stimulatory molecules.
In the 3rd paragraph in the discussion section the authors should also refer and discuss the mechanism by which the different residues on the CD40L/CD40 interaction generates this differential production of pro-inflammatory cytokines, NO induction and expression of PD-L1 on macrophages as shown using mice models in PMID: 32827854. Also, in the 2nd paragraph of the discussion section the authors mention about the dose of anti-CD40. The authors should also comment about the quality of the CD40 stimulation (PMID: 32827854 and PMID: 37596136) as the CD40 receptor has pleiotropic effects and can be modulated for differential function in the future by targeting specific residues of the CD40L-CD40 interaction.
Overall, the work done by Moreau et.al is commendable.
Author Response
The manuscript entitled “Investigating the Use of a Liquid Immunogenic Fiducial Eluter Biomaterial in Cervical Cancer Treatment” submitted by Moreau et.al. discuss the use of a novel multifunctional radiotherapy biomaterial like liquid immunogenic fiducial eluter for the treatment of cervical cancer during radiotherapy. This technique makes use of both the CT and MRI which helps in the guided therapy. The authors showed that the use of this technology helps in controlling the cervical cancer also increasing the survival rates.
The authors have a clear hypothesis and to test this the author used appropriate experimental design. The manuscript is well written, and the statistical methods used are appropriate to the best of my knowledge. The material and methods sections provide complete information. The discussion is well written, highlighting the limitations of the study and shows the future perspective. However, there are some concerns as follows.
The authors need to mention the exact clone of anti-CD40 antibody used as there are different clones available in the market and how was this generated if they are generating it inhouse under the material and methods section.
The clone information for anti-CD40 antibody has been added in Lines 106 – 107.
What was the purpose of using the anti-PD1 antibody as the survival rates were almost same in this group and the CD40 treated group? Also, the authors need to mention the anti-PD1 antibody clone used.
The purpose of using anti-PD1 antibody has been mentioned in Lines 73 – 74. The anti-PD1 clone information has been added in Lines 108 – 109.
The authors have not tested for the release of the cytokines and surface expression of the different inhibitory / stimulatory molecules.
This study focused on any possible efficacy of the LIFE Biomaterial on cervical cancer by assessing tumor volume and mice survival over time. A sentence has been added in the discussion (lines 526 - 527) to indicate that the results motivate future studies exploring the release of cytokines and any surface expression of the different inhibitory/stimulatory molecules.
In the 3rd paragraph in the discussion section the authors should also refer and discuss the mechanism by which the different residues on the CD40L/CD40 interaction generates this differential production of pro-inflammatory cytokines, NO induction and expression of PD-L1 on macrophages as shown using mice models in PMID: 32827854.
Discussion has been added accordingly including the citation.
Also, in the 2nd paragraph of the discussion section the authors mention about the dose of anti-CD40.
Yes, the dose is mentioned in the context of opportunity to vary the anti-CD40 dosing to further optimize treatment outcomes.
The authors should also comment about the quality of the CD40 stimulation (PMID: 32827854 and PMID: 37596136) as the CD40 receptor has pleiotropic effects and can be modulated for differential function in the future by targeting specific residues of the CD40L-CD40 interaction.
Discussion has been added accordingly including the citations.
Round 2
Reviewer 3 Report
Comments and Suggestions for Authors
Great work